

# The seniority quantum number in Tensor Network States

**Klaas Gunst[1⋆], Dimitri Van Neck[1],**
**Peter Andreas Limacher[2] and Stijn De Baerdemacker[3]**

**1** Ghent University, Ghent, Belgium
**2** SAP Security Research, Karlsruhe, Germany
**3** Department of Chemistry, University of New Brunswick, Fredericton, Canada

⋆ klaasgunst@gmail.com

## Abstract

We employ tensor network methods for the study of the seniority quantum number – defined as the number of unpaired electrons in a many-body wave function – in molecular systems. Seniority-zero methods recently emerged as promising candidates to treat strong static correlations in molecular systems, but are prone to deficiencies related to dynamical correlation and dispersion. We systematically resolve these deficiencies by increasing the allowed seniority number using tensor network methods. In particular, we investigate the number of unpaired electrons needed to correctly describe the binding of the neon and nitrogen dimer and the $D_{6h}$ symmetry of benzene.



# 1 Introduction

The quantum mechanical characterization of molecular systems is highly nontrivial due to its many-body character. The need for approximate methods arises for all but the smallest problems. When choosing a suitable approximate method, a consideration has to be made between the accuracy and the complexity of the method. The well-known Hartree-Fock (HF) method provides a mean-field solution for molecular systems rather cheaply. The deficit between the exact ground state energy and the approximate Hartree-Fock energy is an important quantity in quantum chemistry and is called the correlation energy. Often, the distinction between strong (static and nondynamical) and weak (dynamical) correlation is made [1–4]. Although both strong and weak correlations are electronic by nature, they have a different origin; the latter results mainly from the dynamical short-range correlations of electrons, whereas static correlation originates from near-degeneracies of several rivaling electron configurations. Many approximate methods often only excel in capturing one type of correlation. For example, the complete active space self consistent field method (CASSCF) [5,6] and density matrix renormalization group (DMRG) [7–9] are capable of capturing strong correlations within a chosen active space, while coupled cluster (CC) [2,10] and perturbative methods [11] are more suitable for dynamical correlations. In an effort to capture both types of correlation, combinations of these methods have also been developed such as CASPT2 (CAS with perturbation theory up to second order) [12,13], DMRG-CASPT2 [14,15], DMRG-NEVPT2 (DMRG with second-order N-electron valence state perturbation theory) [16], p-DMRG [17], MRCC (multireference coupled cluster) [18–20] and DMRG-TCC (DMRG-tailored coupled cluster) [21–23].

The majority of contemporary electronic structure methods start from a reference state, typically the single-reference HF ground state, and systematically build in correlations by considering elementary excitations from this reference. The conventional approach is to consider particle-hole (ph) excitations from the HF ground state, as is common in CC [2] or truncated configuration interaction (CI) methods [1]. This way, it is possible to construct a hierarchy of multiple $n$-ph excitations which are assumed to be decreasing in importance with increasing $n$. Although tailor-made for dynamical correlations, e.g. in CC theory, it is impractical for static (or non-dynamical) correlation. It was recently observed [24] that the *seniority scheme* is much better suited to capture static correlations associated with the entanglement structure of single-bond breaking processes. Defined as the number of unpaired electrons in a Slater determinant, the seniority quantum number organizes the Hilbert space by the amount of broken closed-shell singlet pairs with respect to a set of (doubly degenerate) spin orbitals. For molecular systems dominated by singlet-pairs bond structures, it was shown that most of the strong static correlation in a system can already be captured in the subspace spanned by all determinants with zero seniority (no unpaired electrons) [24–30]. Although this tremendously reduces the dimension of the Hilbert space at hand, finding the exact doubly occupied configuration interaction (DOCI) wave function is still an exponentially scaling problem. At first glance, the seniority scheme seems only marginally more manageable than the full problem. Interestingly, the antisymmetric product of one-reference orbital geminals (AP1roG) [28,31–33], also known as pair-coupled cluster doubles (pCCD) [34–37], appears to provide a reliable approximation to the DOCI ground state energy for a wide range of molecular systems [36] while staying computationally tractable at a mean-field scaling computational cost [28,38].

Notwithstanding its salient features, there remain several challenges that need to be overcome in order to make the AP1roG wave function quantitatively accurate. The outstanding challenges, which are shared by all methods expressed in the seniority scheme, are (i) the incorporation of dynamical correlation and (ii) the choice of a preferential orbital set, also referred to as the orbital optimization (OO) problem. Another challenge (iii) is the apparent lack of London dispersion correlations in the seniority-zero methods which are crucial to model

large-size molecular systems.

The lack of dynamical correlation in the zero seniority wave functions is well illustrated by the poor description of the correlation energy of the Ne atom, as well as the near-constant parallelity error in the bond dissociation curve of the nitrogen dimer [28, 39, 40]. Dynamical correlation is generally encoded in a set of Slater Determinants with a few ph excitations from the HF reference state. Consequently, methods targeting these excitations, such as (multi-reference) perturbation theory [39, 41], linearized CC [35, 42], extended random phase approximiation (ERPA) [43] or selected configuration interaction (CI) [44], are very well equipped to capture those correlations. However, systematic generalizations of these methods in order to include dynamical correlation from higher-order ph excitations prove either technically and computationally demanding, or break the size consistency of the reference AP1roG wave function.

The concept of paired orbitals is dependent on the choice of the basis orbitals [45]. Current optimization methods unfortunately result into a single unique set of spin orbitals, which can lead to nonphysical symmetry breaking effects in resonating bond structures, such as the aromatic structures in benzene [33], or incorrect characterizations of covalent triplet-bond couplings, such as in the nitrogen dimer [46].

Regardless the correct description of the static correlations associated with bond-dissociation processes, seniority-zero methods have recently been identified as essentially free from London dispersion energy [47], which is remarkable given that 2-electron systems are exactly described by (orbital optimized) seniority-zero methods, capturing the non-covalent Lennard-Jones $1/R^6$ behavior of the dispersion energy in the large $R \rightarrow \infty$ separation limit of the hydrogen dimer.

In order to obtain a global understanding of the deficiencies of the seniority-zero methods, it is quintessential to include all possible broken-pair excitations from higher seniority sectors in a systematic way. Higher seniority subspaces have been studied in the past years using CI approaches, [24, 29, 34, 48, 49], or energy renormalization group (ERG) approaches [50]. The limiting factor of these methods is the pernicious computational scaling whenever no truncation in the Slater determinants is considered. While dynamical correlation is typically included with just a few ph excitations from the HF reference state, corresponding to low-seniority quantum states, it is not clear at present how many broken pairs are needed to restore the correct symmetries or include London dispersion. As a result, there is a need for an analytic method that can assess seniority non-zero contributions in a systematic way at a favorable computational scaling.

In this paper, we use the concept of seniority in junction with tensor network states. In contrast to many other quantum many-body methods, tensor network states consider the whole collection of Slater determinants, and approximate the exact quantum states by restricting the amount of entanglement between local degrees of freedom. Tensor network states are capable of encoding local symmetries of quantum states [51]; therefore they provide a good framework to investigate broken pair excitations, as seniority can be related to the irrep label of the $su(2)$ quasi-spin algebra [52]. In practice, the idea is to perform DMRG in a subspace of the Hilbert space up to a fixed global seniority quantum number, and increase the seniority quantum number until full convergence of the correlation energy is obtained. This procedure will be explained in detail in Section 2. In the proceeding sections, we will present results for the nitrogen dimer (Section 3.1), benzene (Section 3.2) and the neon dimer (Section 3.3), to discuss higher-seniority properties of dynamical correlation, symmetry breaking/restoration and dispersion respectively.

# 2 Methodology

## 2.1 Tensor networks

Pioneered by Steve White in 1992 [7, 8], tensor networks have proven to be a natural language for the entanglement in strongly correlated many body systems. In the tensor network state, each tensor represents a 'local' physical degree of freedom. By connecting them in a network, correlations between the different physical degrees of freedom can be encoded through their virtual degrees of freedom. The exact layout of the network influences the entanglement structure that can be represented; it is easier to correlate physical degrees of freedom that are close in the network.

First and foremost, these tensor network methods have established themselves in the field of condensed matter physics as a wide range of successful tensor networks have been developed for numerous problems. Some notable examples are matrix product states (MPS) [7, 8, 53–55], projected entangled pair states (PEPS) [56] and the multiscale entanglement renormalization ansatz (MERA) [57]. They all provide, in their own way, an efficient representation of certain entanglement structures.

In quantum chemistry, tensor networks have also proven their worth in the study of molecules with strong correlations [9, 58–67]. Quantum chemists don't traditionally study molecules in a Hilbert space built from completely local basis functions (e.g. a grid in three dimensions), but atomic orbital basis sets such as Gaussian-type or Slater-type orbitals are used. These sets give electrons the right flexibility needed for chemistry while the basis size is kept small. On the flip side, the loss of locality in the basis functions makes a suitable network for the entanglement between the physical degrees of freedom less straightforward than for most condensed matter problems. Furthermore, in an atomic orbital basis set, the long range two-body coulomb interactions in the Hamiltonian become four-point interactions. The loss of locality and the need for an efficient evaluation of the Hamiltonian has ensured that the most simple networks are still the most preferred ones. The density matrix renormalization group (DMRG) is, by far, the most popular tensor network method in quantum chemistry and corresponds with the optimization of the linear MPS. Another option for a simple tensor network is the three-legged tree tensor network state (T3NS) [68, 69]. It is a subclass of the more general tree tensor networks (TTNS) [65–67] and was recently introduced by some of us. In this paper, we use these two networks for the study of several chemical systems in restricted seniority subspaces. In the next sections we explain the implementation of restricted seniority for the case of DMRG. However, the ideas are readily adaptable to T3NS and were implemented for both cases in our in-house T3NS-code [70].

## 2.2 Seniority and tensor networks

The non relativistic quantum chemical Hamiltonian to study is given by

$$H = \sum_{ij} t_{ij} \sum_{\sigma} c^\dagger_{i\sigma} c_{j\sigma} + \frac{1}{2} \sum_{ijkl} V_{ijkl} \sum_{\sigma\tau} c^\dagger_{i\sigma} c^\dagger_{j\tau} c_{l\tau} c_{k\sigma} \,, \tag{1}$$

where $i, j, k$ and $l$ are the indices of the orbitals and $\sigma$ and $\tau$ index the spin of the electrons. This Hamiltonian showcases several symmetries, e.g. the particle conservation and total spin symmetry of the electrons. These symmetries can be easily exploited in tensor networks by writing the different tensors in an invariant form under group action of the symmetry [62, 69, 71–78]. Although the seniority is not a symmetry of the quantum chemical Hamiltonian, it is still possible to apply the same idea. In this case, we write each tensor in the network in an invariant form for the seniority. For example, the tensors of rank three present in the MPS can

be made invariant by imposing the following restriction for the tensor elements:

$$T_{a,b,c} = 0, \qquad\qquad \text{if } \nu_a + \nu_b \neq \nu_c , \qquad\qquad (2)$$

or graphically

$$\xrightarrow{\quad} \overset{\downarrow b}{\underset{a}{\widehat{T}}} \xrightarrow{\quad}_{c} = 0, \qquad\qquad \text{if } \nu_a + \nu_b \neq \nu_c , \qquad\qquad (3)$$

where $a$, $b$ and $c$ denote the (physical or virtual) degrees of freedom of $T$ and $\nu_a$, $\nu_b$ and $\nu_c$ are their respective seniority numbers which is well-defined; each state in the degrees of freedom $a$, $b$, and $c$ are eigenstates of the seniority operator. In this example, the seniority number of the first two degrees of freedom of the tensor sum up to the seniority number of the third one. This reflects the fact that seniority is an additive feature, as unpaired electrons from different orbitals all contribute to the total seniority of the state. It is clear that this restriction implies a kind of *flow* for the seniority number in the network which is indicated in Eq. (3) by the directed edges. An example of an MPS wave function built from three of these invariant tensors with the flow indicated is given by

$$|\Psi\rangle = \sum_{\alpha,\beta,a,b,c,f} A_{\text{vac},a,\alpha} B_{\alpha,b,\beta} C_{\beta,c,f} |abc\rangle \qquad\qquad (4)$$

$$= \sum_{a,b,c,f} \text{vac} \xrightarrow{} \overset{\downarrow a}{\widehat{A}} \xrightarrow{}_{\alpha} \overset{\downarrow b}{\widehat{B}} \xrightarrow{}_{\beta} \overset{\downarrow c}{\widehat{C}} \xrightarrow{}_{f} |abc\rangle \qquad\qquad (5)$$

$$= \sum_{f} |\phi_f\rangle , \qquad\qquad (6)$$

where

$$|\phi_f\rangle = \sum_{a,b,c} \text{vac} \xrightarrow{} \overset{\downarrow a}{\widehat{A}} \xrightarrow{}_{\alpha} \overset{\downarrow b}{\widehat{B}} \xrightarrow{}_{\beta} \overset{\downarrow c}{\widehat{C}} \xrightarrow{}_{f} |abc\rangle . \qquad\qquad (7)$$

The physical degrees of freedom (the occupancies of the spatial orbitals) are denoted by $a$, $b$ and $c$ in this example and have a seniority $\nu \in \{0, 1\}$. $\alpha$ and $\beta$ are virtual degrees of freedom. The vacuum state enters the MPS at the leftmost degree of freedom (vac) and has a seniority $\nu = 0$. The final degree of freedom $f$ represents different parts of the total wave function $|\Psi\rangle$. The restriction on the tensors given in Eq. (3) ensures that each part $|\phi_f\rangle$ given by Eq. (7) has a well-defined seniority number, i.e. each Slater determinant $|abc\rangle$ with a non-zero contribution for a particular $|\phi_f\rangle$ has the same seniority number. We can also easily ensure that each $|\phi_f\rangle$ has a unique seniority number by summing any $|\phi_f\rangle$ states with matching seniority numbers. This results in an orthogonal set of $|\phi_f\rangle$ (but not orthonormal). The graphical depiction implies for each connected edge a summation over its corresponding indices; the summation over $\alpha$ and $\beta$ are implied in Eq. (5). This graphical notation is widely used in the tensor network language [51, 56, 69, 79–81].

The only difference with implementing a U(1)-symmetry of the system, e.g. particle conservation or conservation of the spin projection, is the needed summation over the states of the final edge $f$ in Eq. (5). This is necessary as the seniority is *not* a conserved quantum number. Eigenstates of the Hamiltonian are not necessarily eigenstates of the seniority operator and the target state can be a linear combination of Slater determinants with different seniority numbers. To target such a state, the final states at edge $f$ are a set of eigenstates of the seniority operator which combine to the targeted state when summed. In contrast, for a conserved quantity of the system such as the particle conservation, there is only one state $|\phi_f\rangle$ needed.

The set of possible seniority numbers for the wave function is

$$
\Omega = \left\{ n \in \mathbb{N} : \begin{array}{c} n \bmod 2 = N_{\text{tot}} \bmod 2 \\ \left| N_\uparrow - N_\downarrow \right| \le n \le \min\left(N_{\text{tot}}, 2k - N_{\text{tot}}\right) \end{array} \right\} , \tag{8}
$$

with $k$ the number of spatial orbitals, $N_\uparrow$ ($N_\downarrow$) the number of electrons with spin up (down) and $N_{\text{tot}}$ the total number of electrons. For every renormalized state in the last edge, we have $\nu_f \in \Omega$. By restricting $\nu_f$ to a subset $S$, i.e. $\nu_f \in S \subseteq \Omega$, ground states in seniority-restricted subspaces can be targeted. The weight of each seniority subspace for the total wave function can be readily calculated as $|c_{\nu_f}|^2 = \langle \phi_f | \phi_f \rangle$.

In a similar fashion, one could also use other non-conserved quantum numbers than the seniority. For example we could use the excitation number with respect to the Hartree Fock wave function. By only allowing Slater determinants with a certain amount of excitations, tensor networks can be used as an approximate configuration interaction (CI) solver with arbitrary allowed excitation levels.

### 2.2.1 Suboptimal decomposition

When using a wave function ansatz as shown in Eq. (5), we impose a restriction on the left renormalized states at each splitting of the network. Due to the fact that a vacuum state enters in the left most edge (vac) and all tensors used are invariant under the seniority operator, the left renormalized states need to have a well defined seniority number. This restriction does not hold for the right renormalized states since multiple states with different seniority exit at the right most edge $f$.

This restriction results in the need of a possibly larger bond dimension than when discarding seniority. We illustrate this using a wave function with three electrons in three spatial orbitals:

$$
|\Psi\rangle = \frac{1}{\sqrt{2}} \left[ |\uparrow, \downarrow\rangle \otimes |\uparrow\rangle + |-, \uparrow\downarrow\rangle \otimes |\uparrow\rangle \right] \tag{9}
$$

$$
= \frac{1}{\sqrt{2}} \left[ |\uparrow, \downarrow\rangle + |-, \uparrow\downarrow\rangle \right] \otimes |\uparrow\rangle . \tag{10}
$$

In Eq. (10), the Schmidt decomposition for a partitioning between the first two and the last orbital is given. At this partitioning only a virtual bond dimension of one is needed to represent the state. However, when we impose that the left states, i.e. the states in the first two orbitals, of the decomposition should also be eigenstates of the seniority operator, the needed bond dimension at this partitioning increases to two, confer Eq. (9).

## 2.3 DOCI and tensor networks

Restricting the calculation to configurations with $\nu = 0$, i.e. all electrons are paired, is easily done with the aforementioned method. However, it is more efficient to directly implement the quantum chemical Hamiltonian projected on the DOCI-subspace where only paired electrons are allowed. The DOCI-Hamiltonian is given by

$$
H_{\text{DOCI}} = 2 \sum_i t_{ii} n_i + \sum_{ij} \left( 2V_{ijij} - V_{ijji} \right) n_i n_j + \sum_{i \ne j} V_{iijj} b_i^\dagger b_j , \tag{11}
$$

where $b_i^\dagger$ and $b_i$ are the bosonic pair creation and annihilation operators and $n_i$ is the pair number operator at orbital $i$. They are given by

$$
b_i^\dagger = c_{i\uparrow}^\dagger c_{i\downarrow}^\dagger , \qquad\qquad b_i = c_{i\downarrow} c_{i\uparrow} \tag{12}
$$

and

$$n_i = b_i^\dagger b_i \ .$$ (13)

This Hamiltonian only scales quadratically with the number of orbitals in contrast with the quartic scaling of the full Hamiltonian. TNS calculations in the DOCI subspace can be performed with a lower polynomial scaling, as stated in Table 1 for the DMRG and T3NS.

Table 1: Resource requirements for DMRG and T3NS with renormalized operators for the full quantum chemical Hamiltonian (Eq. (1)) or the DOCI Hamiltonian (Eq. (11)) without orbital optimization for $k$ spatial orbitals. The maximal virtual bond dimension is denoted by $D$.

|  | CPU time | Memory |
|---|---|---|
| QC-DMRG: | $\mathcal{O}\left(k^4D^2 + k^3D^3\right)$ | $\mathcal{O}\left(k^3D^2\right)$ |
| DOCI-DMRG: | $\mathcal{O}\left(k^2D^3\right)$ | $\mathcal{O}\left(k^2D^2\right)$ |
| QC-T3NS: | $\mathcal{O}\left(k^4D^2 + k^3D^4\right)$ | $\mathcal{O}\left(k^3D^2 + kD^3\right)$ |
| DOCI-T3NS: | $\mathcal{O}\left(k^2D^4\right)$ | $\mathcal{O}\left(k^2D^2 + kD^3\right)$ |

We find that DOCI ground state wave functions have in general lower entanglement than their corresponding full configuration interaction (FCI) ground state wave function; accurate results for DOCI can be obtained with a much lower bond dimension. The synergy between the lower polynomial scaling and the lower bond dimension needed, makes DOCI-TNS very fast and a good option for initializing tensor network calculations in the FCI space. For example, DOCI calculations without orbital optimization with 162 electron pairs and 261 spatial orbitals can be executed in a several minutes on a common laptop [82].

Not only DOCI-TNS but also general seniority-restricted tensor network calculations can provide interesting approximations from a computational viewpoint, although they have the same polynomial scaling as unrestricted calculations. They can converge faster than the latter due to the lower entanglement present in the wave function; hence a lower bond dimension is needed. This is generally the case for $\nu \leq 2$ and (to a lesser extent) $\nu \leq 4$ calculations.[1]

In contrast, using high seniorities will result in a loss of efficiency when compared to seniority-unrestricted tensor network calculations due to the extra bookkeeping needed and the suboptimal Schmidt decomposition (see Section 2.2.1). Especially the latter is detrimental as the sought-after wave function at high seniority calculations is similarly entangled as the exact solution, but the tensor network ansatz can not capture it as efficiently as the equivalent unrestricted ansatz. As such, these types of calculations primarily provide a means to analyze the need for broken pairs in chemical systems. They should not necessarily be viewed as an efficient way to approximate the FCI solution. The presented tensor network method allows to investigate the number of broken pairs needed for recovering a qualitative correct picture when one attempts to correct existing seniority-zero methods.

## 3  Applications

We discuss some calculations with the seniority-restricted tensor network code. As these calculations are orbital dependent, several types of orbitals are considered. The effect of allowing progressively more broken pairs is also studied within each orbital set. In Section 3.1 and Section 3.3, the dissociation of the nitrogen and neon dimer are considered, respectively.

---

[1]Bear in mind that, while these calculations converge faster, they do not converge to the FCI limit.

Section 3.2 discusses the benzene molecule, a system demonstrating artificial $D_{6h}$ symmetry breaking in the seniority-zero subspace [33].

Coupled cluster natural orbitals and Löwdin orthogonalized atomic orbitals are obtained with PySCF [83–85]. DOCI-optimized orbitals are generated through an in-house DOSCF code and were carefully checked to correspond to the lowest possible DOCI energy, i.e. the global minimum [46]. The seniority-restricted tensor network calculations were executed with our own T3NS-code [70]. All seniority-restricted tensor network calculations are MPS calculations. We exploit the spin symmetry and the reported bond dimensions for the tensor networks are *reduced* bond dimensions; renormalized states belonging to the same multiplet are represented by one reduced renormalized state, thus reducing the needed bond dimension. Seniority-restricted tensor network calculations are, just as regular tensor network calculations, not exact; the accuracy can be controlled by the bond dimension. The following calculations use a large enough bond dimension to ensure quantitatively accurate potential energy surfaces.

## 3.1  Nitrogen dimer

Characterized by a triple bond breaking, the nitrogen dimer is a much visited test case for new quantum chemical methods, and has already been investigated as such in the seniority framework by Bytautas *et al.* [24] using an active space in the cc-pVDZ basis with $D_{2h}$-symmetry adapted MOs. Here, we study the nitrogen dimer in a cc-pVDZ basis set with all electrons correlated, however the DMRG results are qualitatively similar to the results in [24]. Seniority-restricted spin-adapted DMRG with a reduced bond dimension up to a 1000 is used to optimize the ground state in the different subspaces. The allowed seniority increases from 0 (DOCI) up to 10 for the largest calculations, allowing 5 electron pairs to be broken. In Fig. 1, the dissociation curves are given for calculations within the different seniority subspaces. Calculations were performed for canonical orbitals (Fig. 1a), DOCI-optimized orbitals (Fig. 1b) and CCSD natural orbitals (Fig. 1c). Although the DOCI-optimized orbitals are optimized for the seniority-zero subspace specifically they also perform better in higher seniority subspaces, albeit marginally. Eventually for $\nu \leq 8$ and onward, all orbital sets give quasi-FCI energies.

In Ref. [32], it is shown that the seniority-two sector decouples from the seniority-two-plus-zero sector up to first order for DOCI-optimized orbitals; only a small correction should occur due to the introduction of single broken pairs in this orbital set. Putting this first order decoupling to the test, we notice indeed a small energy correction for the DOCI-optimized orbitals, smaller than for canonical orbitals. In Fig. 2, the weights of the different seniority subspaces are plotted for the ground state in both canonical and DOCI-optimized orbitals. It is yet another illustration that for DOCI-optimized orbitals (Fig. 2b) the seniority-two subspace is less important than for canonical orbitals (Fig. 2a). However, a first order decoupling is not an exact one; there are other orbital sets possible which give even smaller energy corrections. This is illustrated by the natural orbitals (Fig. 1c) which give even smaller energy corrections when allowing single broken pairs in this system.

As a last observation we note that the largest change in energy occurs when including the seniority-four subspace, and this for all orbital sets in Fig. 1. This trend was also noticed in Ref. [24] for the nitrogen dimer in nonlocal orbitals. When including up to seniority four the energies are close to converged around the binding distance for increasing seniority numbers; however, the binding energy itself is still overestimated due to missing dynamical correlation at the dissociation (values are given in Table 2 for both canonical and DOCI-optimized orbitals).

Intuitively, we would expect a much larger error when excluding the seniority-six subspace as Hund's rule dictates dissociation to two nitrogen atoms with each three unpaired electrons. However, seniority and pairing is an orbital-dependent concept [46]; we need to keep in mind that Hund's rule applies to a nitrogen atom with orbitals localized around that atom. To study the interpretation of Hund's rule in non-local orbitals, we consider a toy model of two sets

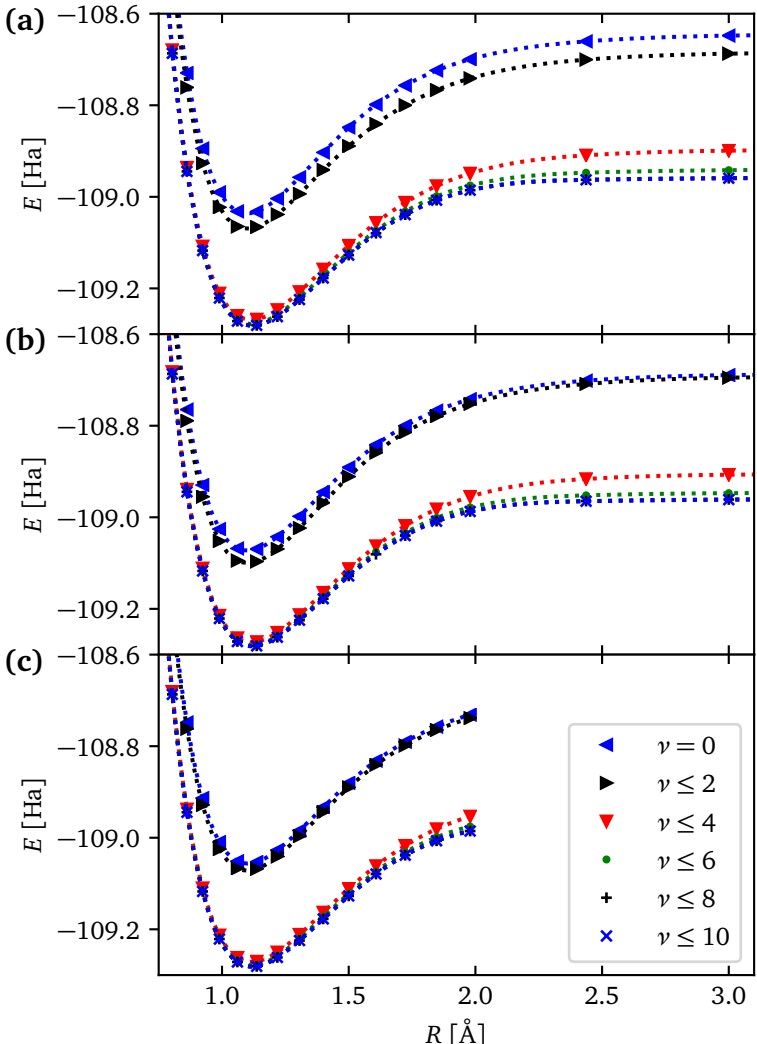

Figure 1: Dissociation curves at different seniority subspaces for the nitrogen dimer. Results for canonical orbitals (a), DOCI-optimized orbitals (b) and CCSD natural orbitals (c) are given. For (c), only results where CCSD converged are plotted.

of three orbitals $(p_x, p_y, p_z)$ and $(p'_x, p'_y, p'_z)$. Each set of orbitals mimics the local $p$-orbitals of each nitrogen atom which are singly occupied and couple together to a $S = 3/2$ state, as dictated by Hund's rule. Our tensor network calculations target over the whole dissociation curve a singlet state for the dimer, so the two toy-nitrogen atoms should couple as $[3/2, 3/2]^0$. Mimicking non-local orbitals, we rotate the orbitals pairwise as follows:

$$\pi_1 = p_x \cos\theta + p'_x \sin\theta, \qquad \pi_1^* = -p_x \sin\theta + p'_x \cos\theta$$
$$\pi_2 = p_y \cos\theta + p'_y \sin\theta, \qquad \pi_2^* = -p_y \sin\theta + p'_y \cos\theta$$
$$\sigma = p_z \cos\theta + p'_z \sin\theta, \qquad \sigma^* = -p_z \sin\theta + p'_z \cos\theta \ .$$

In Fig. 3, the weights of the different seniority sectors is given for the $[3/2, 3/2]^0$ coupled toy wave function in function of the rotation angle $\theta$. As can be seen in this model seniority-six is actually of no importance when working with delocalized orbitals ($\theta = \pi/4$). Instead, the correct dissociation can be described with only seniority-zero-plus-four and both seniorities equally important.

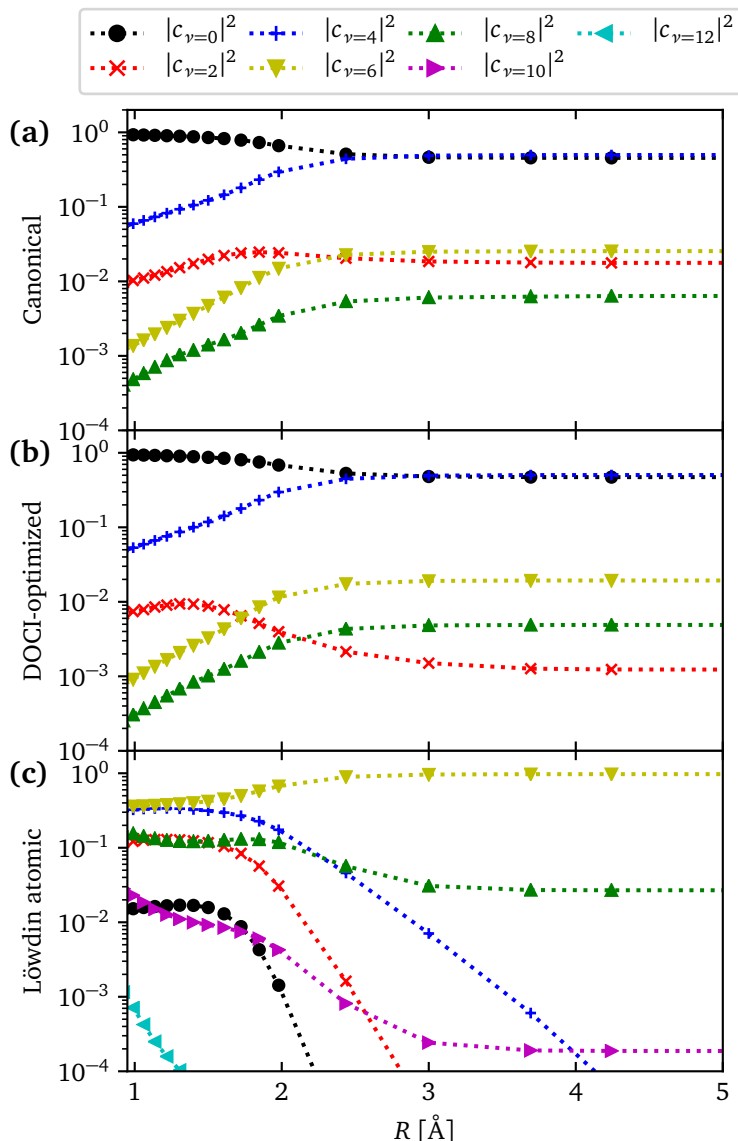

Figure 2: Weights of the different seniority subspaces for the ground state wave function of the nitrogen dimer. Results for canonical orbitals (a), DOCI-optimized orbitals (b) and Löwdin orthogonalized atomic orbitals (c) are given.

Both canonical orbitals and the DOCI-optimized orbitals are delocalized for the $2p$-orbitals in this system. This dominating importance of the seniority zero and four for the wave function at dissociation is very clear in Fig. 2a and Fig. 2b. The other seniority sectors have very small contributions in comparison. As an illustration, we also included calculations with Löwdin orthogonalized atomic orbitals in Fig. 2c. As these orbitals are localized, it corresponds with $\theta = 0$ in Fig. 3. These orbitals do give rise to a very important seniority-six subspace at dissociation, as predicted by Hund's rule. Evenmore, all seniority sectors smaller than six express a superexponential decay.

## 3.2 Benzene

In this section, the in-plane distortion of benzene is studied. The exact nature of the distortion is given in the inset in Fig. 4 and is characterized by the angle $\theta$. At $\theta = 60°$ the $D_{6h}$ symmetric

Table 2: Binding energies in m$E_\mathrm{h}$ for seniority-restricted calculations in both canonical and DOCI-optimized orbitals.

|  | $\nu = 0$ | $\nu \leq 2$ | $\nu \leq 4$ | $\nu \leq 6$ | $\nu \leq 8$ | $\nu \leq 10$ |
|---|---|---|---|---|---|---|
| canonical | 424 | 382 | 377 | 338 | 322 | 322 |
| DOCI-optimized | 383 | 405 | 367 | 333 | 321 | 321 |

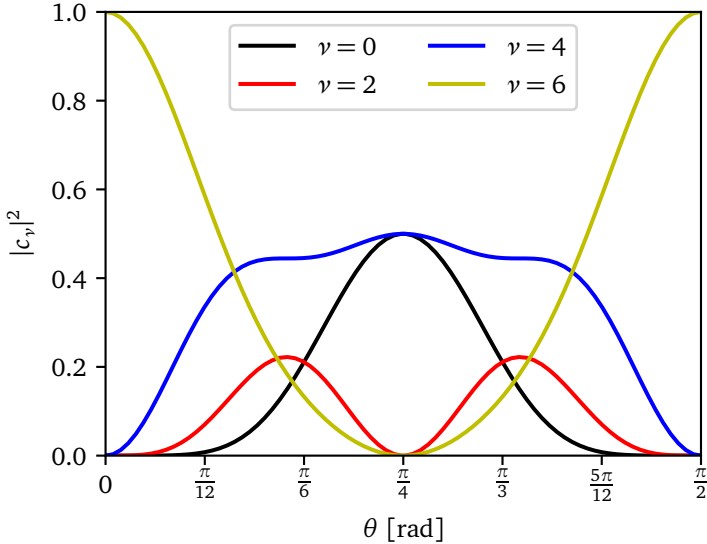

Figure 3: Toy model of two nitrogen atoms with both $S = 3/2$ respecting Hund's rule. The two atoms couple together to a singlet. The figure represents the weights of the different seniority sectors for local $(0, \pi/2)$, delocalized $(\pi/4)$ orbitals and everything in between.

equilibrium structure geometry of benzene is obtained. At other angles, the distortion introduces alternating shorter and longer carbon-carbon bonds. For this system Boguslawski *et al.* [33] showed that benzene ($\theta = 60°$) is not the equilibrium structure within the seniority-zero subspace; an artificial symmetry breaking occurs when allowing orbital optimization. In this paper we use the experimental geometry of benzene [86,87] and distort the angle while keeping the atomic distances to the center of mass intact; we did not perform a geometry optimization while distorting the angle.

We use DOCI-optimized orbitals in the STO-6G basis set to study this artificial symmetry breaking with all electrons correlated. We chose STO-6G as the distortion angle of the minimal energy DOCI structure is particularly large for this basis set. The tensor network calculations are executed with a reduced bond dimension of 1000.

In Fig. 4 the results for the ground state in the different seniority subspaces are given. In accordance with Boguslawski *et al.* [33], we notice that, indeed, the ground state is not found at $\theta = 60°$ in the seniority zero subspace. When the breaking of one pair is allowed in this orbital set, the correction is rather small and the correct symmetry is not restored; as expected due to the aforementioned first order decoupling of the seniority-zero and seniority-two subspace in these orbitals.

When including progressively higher seniorities, the stable configuration moves closer to the expected $D_{6h}$ symmetric benzene. The potential energy surface enjoys a large qualitative correction when including the seniority-four subspace in the calculations. However the predicted most stable configuration is still off by 0.59°. The inclusion of seniority-six further improves

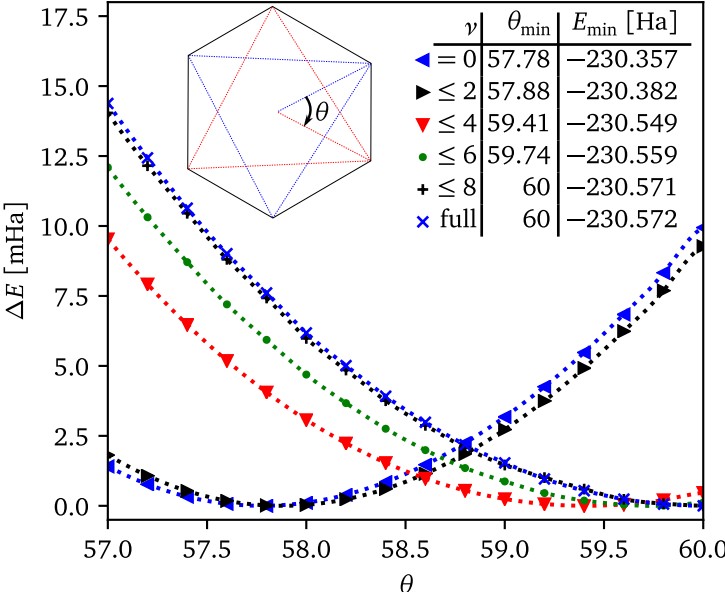

Figure 4: Benzene in a STO-6G basis set for different distortion angles. The minimal energies and the corresponding distortion angles for increasing seniority subspaces are given in the inset. A graphical depiction of the in-plane distortion of the aromatic ring in benzene is also shown.

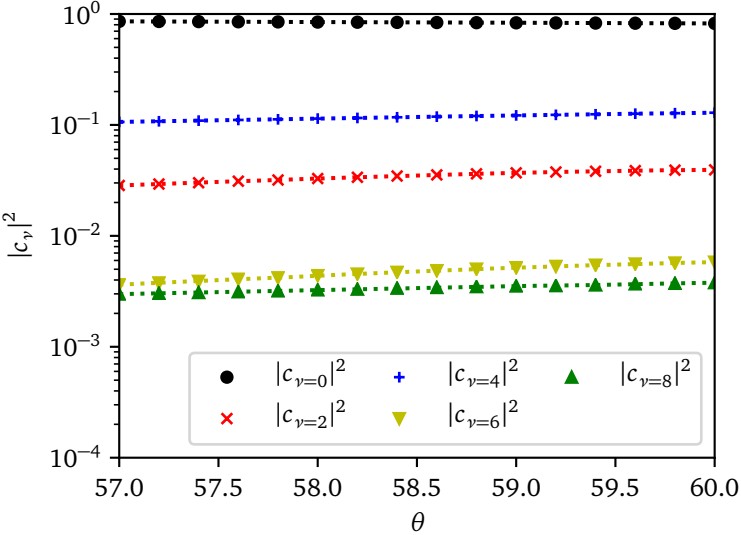

Figure 5: Weights of the different seniority subspaces for the ground state wave function of the in-plane distorted benzene in DOCI-optimized orbitals.

the quality of the potential energy surface, but only at seniority-eight the correct symmetry seems to be recovered, at least up to the resolution of our performed calculations. At this point, the results become very close to the full seniority results. In Fig. 5, the weights of the different seniority sectors in the ground state during distortion are also given. These weights do not express the large changes as were seen during dissociation of the nitrogen dimer in Fig. 2. This is quite expected as the bond breaking of the nitrogen dimer is a far more outspoken change than the small benzene distortions in this section.

## 3.3 Neon dimer

The neon dimer, constituted by just two noble gas atoms, is very weakly bound. Although the electrons do not form covalent bonds between the two atoms, it expresses some bonding character due to weak dispersion forces. In Ref. [88], an empirically fitted potential curve results in a binding energy of $-134\,\mu E_\mathrm{h}$ and a binding distance of $3.091\,\text{Å}$.

As the binding of the neon dimer is rather weak and due to dynamical correlations, it will be very sensitive to the chosen basis set size. For an accurate description of the potential energy curve, a large basis set should be chosen and basis set superposition errors (BSSE) should be taken into account appropriately [89]. A clear example of the importance of BSSE-corrections is the dissociation curve on the Hartree-Fock level. At this level of theory no binding is expected as the Hartree-Fock solution is dispersion-free. However, when using small basis sets, one would find a binding neon dimer at the Hartree-Fock level if one neglects to correct the BSSE [89].

We study the neon dimer in the aug-cc-pVDZ basis; it was found that this basis set has a favorable tradeoff between mitigating BSSE and numerical stability issues of larger basis sets. Calculations with different seniority sectors are executed while using DOCI-optimized orbitals with a frozen $1s$ core. Reduced bond dimensions up to 800 are used for the DMRG calculations. As the aug-cc-pVDZ basis is a rather small basis for capturing dispersion forces, appropriately removing BSSE is important. This is done by using the Boys and Bernardi counterpoise correction [90].

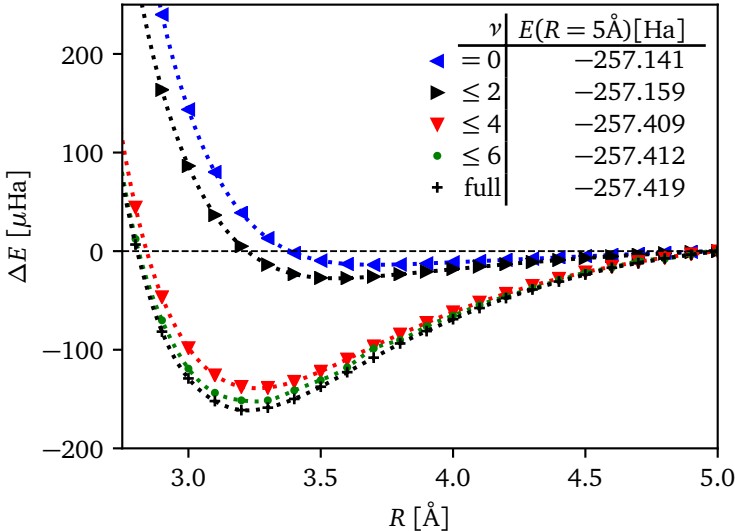

Figure 6: Dissociation curves for the neon dimer without BSSE-correction. The energies at large separation distances is given in the inset.

In Fig. 6 the raw uncorrected results are given for the different calculations. For all seniority calculations the neon dimer seems to be bound. However, for seniority-zero and seniority-two-plus-zero is is very weakly bound; only for $\nu \leq 4$ calculations and higher the neon dimer bounds qualitatively corresponding with the full seniority case. For the counterpoise correction, equivalent calculations as for the dimer are executed but where one neon atom is replaced by a chargeless, electronless ghost atom. This way, we can approximately correct for the extra stabilization each neon monomer experiences by the extra added basis functions of the other monomer. The BSSE-corrected dissociation energy for the dimer at distance $r$ is then given by

$$E_\mathrm{dissoc}(r) = E_\mathrm{Ne-Ne}(r) - E_\mathrm{Ne-ghost}(r) - E_\mathrm{ghost-Ne}(r) . \tag{14}$$

The same level of theory should be used for these ghost calculations as for the original

calculation. This poses a difficulty since the seniority restricted calculations are not size consistent; $E_{\text{dissoc}}(r \to \infty)$ does not tend to zero as is desired. Assume we have executed a dimer calculation with $\nu \leq 4$, using ghost calculations with the same $\nu \leq 4$ will over-correct, while ghost calculations at the lower $\nu \leq 2$ will under-correct. We try to solve this problem by both over- and under-correcting and shift both curves to 0 in the dissociation limit. Results for these BSSE-corrections are given for different seniority sectors in Fig. 7 with the grayed area indicating where the exact BSSE-correction is expected to be. From Fig. 7a and Fig. 7b, it seems that the weak bound present in Fig. 6 for $\nu = 0$ and $\nu \leq 2$ practically or completely disappears when taking BSSE-corrections into account. When correcting $\nu \leq 4$ calculations, the over-corrected dissociation underestimates the dissociation energy a bit with respect to the BSSE-corrected full-seniority tensor network calculations (which approximate FCI) while the under-corrected dissociation overestimates the dissociation energy, as can be expected.

It seems thus that calculations with seniority-zero and seniority-two do not model the needed dispersion and at least seniority-four is needed. Taking into account that the DOCI-optimized orbitals are localized on separate Neon atoms for larger separations, this suggests the breaking of at least one electron pair at each Ne atom is needed, inducing polarization effects in each atom which give rise to the dispersion energy.

Finally, we notice that the full-seniority dissociation energy with BSSE-correction is a factor of three smaller than empirical measurements and the bond length is overestimated. This is quite normal when studying dynamical correlations in small basis sets. The limited basis set does not allow all the needed flexibility for the stabilization of the dimer.

# 4 Conclusion

In this paper, the concept of seniority is joined with tensor network methods. By using seniority-invariant tensors in a tensor network, we can force all the renormalized states in the virtual bonds to be eigenstates of the seniority operator. This allows for arbitrary seniority-restricted calculations. For DOCI (doubly occupied configuration interaction) calculations, we can immediately implement the DOCI-projected quantum chemical Hamiltonian in Eq. (11). This results in a very fast tensor network calculation, partly because of the simpler Hamiltonian, partly because the correlations in the seniority-zero subspace for molecular systems are easily captured by tensor networks; even for very large systems, a bond dimension of less than 100 suffices for energies within chemical accuracy of the exact DOCI energy.

The seniority-restricted tensor network method opens up novel ways for efficient approximate DOCI algorithms with orbital optimization. As one-body and two-body reduced density matrices are easily extracted from the TNS, one could alternate between DOCI-TNS calculations and orbital optimizations by using e.g. Newton-Raphson based algorithms [5] or Jacobi rotations [91, 92]. As in Ref. [93], one could also intertwine the orbital optimization with the tensor network calculation itself. Instead of lowering the entanglement, one could now optimize for the DOCI energy or perform a seniority zero-plus-two calculation and minimize the seniority two contribution by orbital rotations. A proxy for decoupling the seniority-zero and seniority-two subspaces has been previously done in Ref. [32].

Several systems are studied within different seniority subspaces. For the dissociation of the nitrogen dimer, only a quantitative dissociation curve can be obtained when at least two pairs are allowed to be broken. This can be theoretically explained due to Hund's rule. The in-plane distortion of benzene and its artificial $D_{6h}$ symmetry breaking in the seniority-zero subspace [33] is also studied. A large correction of the artificial symmetry breaking occurs when including seniority-four, however up to eight unpaired electrons are needed for a complete restoration of the correct benzene point group symmetry in the used basis set. Finally, also the

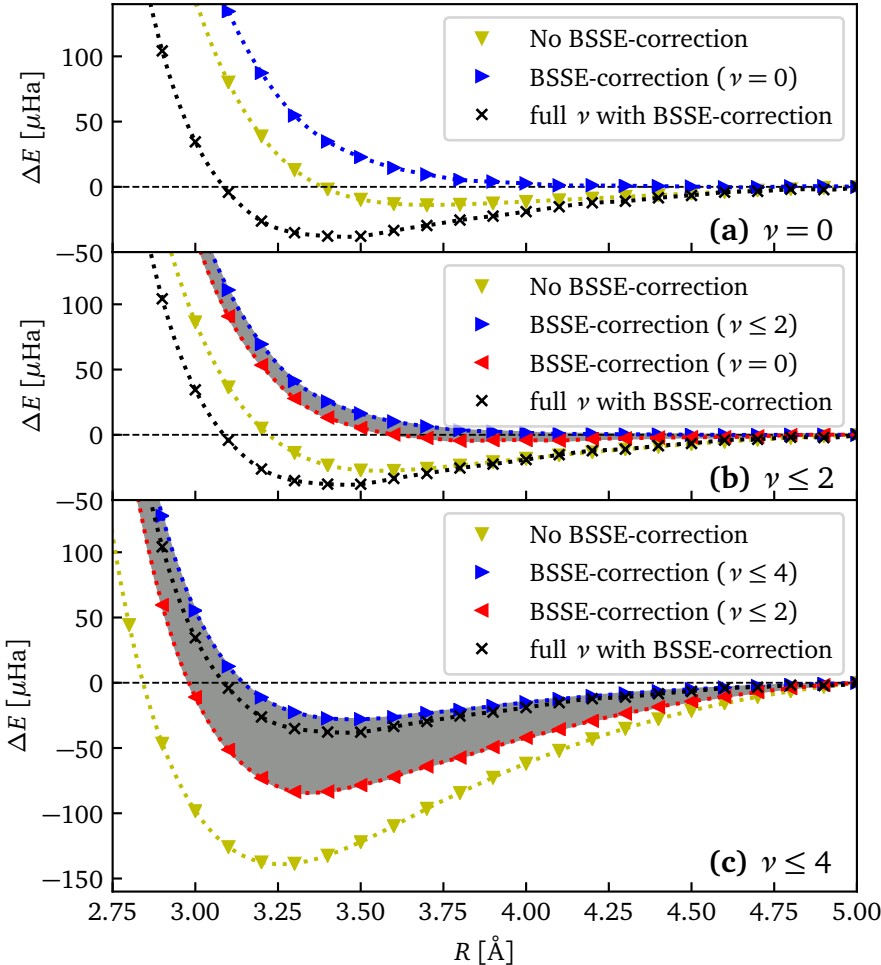

Figure 7: Dissociation curves for the neon dimer with and without BSSE correction for different seniority subspaces. When BSSE-correction is performed, the used seniority subspace for the ghost atom calculation is given in brackets in the legend. The dissociation curve for a full-seniority calculation with full BSSE-correction is also shown in each subfigure. Results are shown for $\nu = 0$ (a), $\nu \leq 2$ (b) and $\nu \leq 4$ (c) subspace calculations.

dissociation of the neon dimer is considered. At the seniority-zero level of theory the neon dimer is non-binding; DOCI does not capture the dispersion forces needed for the weak binding characteristic of neon. Only at seniority-four and onward, the dispersion forces are adequately picked up.

For all systems, the seniority-two subspace has only a small contribution to the total wave function when using DOCI-optimized orbitals; as expected by the theoretical first order decoupling between seniority-zero and seniority-two subspaces in these types of orbitals [32]. However, a first order decoupling is not an exact decoupling and other orbital sets can be found which attribute even less importance to the seniority-two subspace. An example of this is given by the natural orbitals of the nitrogen dimer in Fig. 1c.

## Acknowledgements

KG acknowledges financial support from the Research Foundation Flanders (FWO Vlaanderen). This research was undertaken, in part, thanks to funding from the Canada Research Chairs Program (SDB). Computational resources and services were provided by Ghent University (Stevin Supercomputer Infrastructure), Compute Canada, the organization responsible for digital research infrastructure in Canada, and ACENET, the regional partner in Atlantic Canada. ACENET is funded by the Canada Foundation for Innovation (CFI), and the provinces of New Brunswick, Newfoundland & Labrador, and Nova Scotia. Discussions with Patrick Bultinck are gratefully acknowledged.

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
