# Peer review of "The seniority quantum number in Tensor Network States"

_SciPost Chemistry, doi:SciPost Chem. 1, 001 (2021)_

## Round 1 · Referee Report · Anonymous (Referee 1) · 2020-10-30

Report

Gunst, Van Neck, Limacher, and De Baerdemacker study the convergence of the electronic structure of many-electron systems with seniority number, using a tensor network representation of the wave function. The analysis is clear and the conclusions reflect similar findings in earlier studies. To my knowledge, this work is the first to present a seniority-based analysis involving tensor networks. I particularly like the analysis of the toy model for N2 dissociation, which provides a clear illustration of the orbital dependence of the relevance of different seniority sectors. Overall, I think this is a nice contribution, and I have only minor comments and questions.

Requested changes

  1. Page 2: “For example, the complete active space self consistent field method (CASSCF) and density matrix renormalization group (DMRG) are capable of capturing strong correlations…” DMRG applied within the full space should also account for dynamic correlation, so this statement should be modified to acknowledge that the authors are referring to DMRG in an active space.

  2. Page 2: “Interestingly, the antisymmetric product of one-reference orbital geminals (AP1roG), also known as pair-coupled cluster doubles (pCCD), appears to provide a reliable approximation to the DOCI ground state solution…” I would change “solution” to “energy” since pCCD does not necessarily provide a good description of other properties. For example, right pCCD wave functions can be good approximations to DOCI ones, whereas the left pCCD wave function can be significantly different (JCP 142, 214116 (2015)), meaning that the respective density matrices could differ.

  3. Page 5: it would be nice if the notation in Eq. 4 was clarified.

  4. Page 6: FCI is used for the first time, but the acronym was not defined.

  5. Page 6: “For example, DOCI calculations with 162 electron pairs and 261 spatial orbitals have been executed in a few minutes on a common laptop.” This assertion requires a citation. Also, a clarification would be appreciated: did this calculation involve an orbital optimization?

  6. Page 7: Table 1: It should be clear that the resource requirements do not include the orbital optimization step often performed in DOCI.

  7. Page 9: “When including up to seniority four the energies are close to FCI around the binding distance…” No FCI results are presented for N2, so this statement should be modified to suggest simply that the energy has converged with respect to seniority number by omega = 4. Otherwise, FCI results should be included or a citation provided.

  8. Page 12: “When including progressively higher seniorities, the stable configuration moves closer to the expected D6h symmetric benzene.” How is the stable configuration determined? Is a full geometry optimization performed, or do the authors just scan the angle theta? If they are just scanning an angle, I wonder how the remaining geometric paramaters change. Either way, some additional information would be useful for anyone trying to reproduce this study.

  9. Section 3.3: This is a somewhat silly request, but I would prefer that the authors remove the language about “bonding” in favor of some other phrase (e.g., “is very weakly bonding” -> “is very weakly bound” on page 12, “to be bonding” -> “to be bound” on page 13, etc.) because the favorable interaction in neon dimer can hardly be thought of as a bond.

  10. Page 13: “The over-corrected dissociation underestimates the dissociation energy a bit with respect to the FCI BSSE-corrected calculations…” FCI results are not presented. Do the authors mean to say tensor network calculations with full seniority?

  11. Page 14: “This suggests the breaking of at least one electron pair at each Ne atom is needed, inducing polarization effects in each atom which give rise to the dispersion energy.” This is a reasonable conclusion to draw regarding the importance of seniority four sector, except that the basis used is a delocalized one, correct? Can the physical picture of broken pairs at atomic centers be applied here?

  12. Lastly, it seems to me that the seniority-based expansion is not necessarily good for reducing the overall complexity of a calculation, if one wants quantitative accuracy. Seniority eight is required for exact restoration of symmetry in benzene, and seniority of at least four is required to capture dispersion. This leads me to some questions. First, what is the prospect for generally useful seniority-based expansions of the wave function? Is the best route to stop at seniority zero and then apply some correction (e.g., AP1roG + ERPA)? Second, if a better route is to perform some high-seniority-number (8?) calculation in a tensor network framework, I wonder how the complexity of intermediate seniority calculations compares to the DOCI or full seniority limits in tensor network calculations. Can the authors comment?

  • validity: high
  • significance: high
  • originality: high
  • clarity: top
  • formatting: perfect
  • grammar: perfect

Author:  Klaas Gunst  on 2020-12-15  [id 1082]

(in reply to Report 1 on 2020-10-30)
Category:
answer to question

We would like to thank the referee for his/her thorough reading and the comments made in the report. We have addressed the points raised in the resubmission. We would like to further answer on two questions raised by the referee:
8) We have used the experimental value as given in the NIST database for the distances between the center of mass and the different atoms. We keep these fixed and indeed only variate the angle. We have added this to the manuscript. As to give some insight in the change of the remaining geometric parameters, we have done a more exhaustive scan of the geometries in a restricted active space. We have performed orbital-optimized DOCI calculations in an active space spanned by the 2pz-orbitals of each C atom in the same sto6g basis-set. This very restricted active space of 6 orbitals and 6 electrons exhibits the same symmetry-breaking as the calculations presented in the paper with the exception that seniority 6 restores the symmetry in this case. While these calculations in the restricted active space are not conclusive, they do suggest that the symmetry-breaking does not vanish when optimizing the geometry while distorting. We have attached these results (benzene_min_act.pdf) for the ground state energies in the different seniority subspaces in this small active space. A step-size of 0.05 degrees and 0.005Å for the center-of-mass to carbon distance was used. The carbon-hydrogen distances were also varied and the best distance was used for each point shown in the contour plots.
11) The DOCI-optimized orbital for the Neon dimer are delocalized on one of the two atoms for larger separations. We have clarified this in the manuscript.

Attachment:

benzene_min_act.pdf

---

## Round 1 · Referee Report · Anonymous (Referee 2) · 2020-11-25

Report

This manuscript descibes the optimization of seniority-number
restricted wavefunctions using a tensor network
representation. Overall, the work is very nice and well organized. I
only have minor comments and suggestions.

Requested changes

  1. As the authors say, one critical aspect in DOCI calculations is the choice of the right orbital basis. The authors should comment on the prospect of doing orbital optimization within their tensor-network scheme.

  2. The authors should emphasize more the low-entanglement structure of the DOCI wavefunction. The important consequence of this is that the tensor network can be optimized rather efficiently.

The authors should comment on the entanglement structure once higher seniorities are involved. In particular, if one has to reach relatively high seniorities or if the entanglement structure increases considerably, then it may not be readily apparent why one should use an MPS with restricted seniorities vs the full MPS in DMRG.

  1. A clarification regarding, e.g., Fig 2, would be appreciated. It seems to me that the weights plotted correspond to a splitting of the wfn: |Psi> = c0 |v=0> + c2 |v=2> + ... Yet, this is not described in any Eqn and I assume it can lead to confusion.

  2. While I'm not completely familiar with Ref 44, I cannot agree with the statement

"Size consistency of the AP1roG wave function, or by extension any seniority-zero method, is guaranteed when the spin orbitals are optimized such that the energy is in a variational minimum."

In fact, the optimal DOCI orbitals for N2 are delocalized and therefore the solution is not size consistent.

  • validity: high
  • significance: high
  • originality: top
  • clarity: top
  • formatting: excellent
  • grammar: excellent

Author:  Klaas Gunst  on 2020-12-15  [id 1081]

(in reply to Report 2 on 2020-11-25)
Category:
answer to question

We would like to thank the referee for his/her thorough reading and the comments made in the report. We reply to the comments in the report point-by-point in the following:

  1. We have commented upon the usage of DOCI-TNS in combination with orbital optimization in the conclusions.

  2. We have further commented on the entanglement present in the seniority-restricted wave functions at the end of section 2.

  3. We have elaborated on this in section 2.2.

  4. While the orbital-optimized DOCI wave function is indeed not separable for dinitrogen, this is due to the fact that DOCI is unable to describe (at least without introducing ghost orbitals) a single nitrogen atom as it has an odd amount of electrons. When DOCI is able to describe the constituents of the system, it is size-consistent.

As we agree that it is at least debatable if DOCI is size consistent, we have left out the given sentence in the manuscript. This does not impinge the general message of the paragraph.

---

## Round 2 · Referee Report · Anonymous (Referee 2) · 2020-12-16

Report

The authors have addressed the comments from the reviewers. I therefore recommend publication.

---

## Round 2 · Author Response

We would like to thank the referees for their helpful comments and the thorough reading. We have revised the manuscript taking into account their request for changes.

---

## Round 2 · List of Changes

We list the changes made to address the comments of the referees.

In response to the first referee:

  1. We agree with the remark of the referee. We have modified the statement to reflect active space calculations.

  2. We agree with the remark of the referee, we have changed the statement to only reflect on the accurate energies obtained by AP1roG.

  3. We have further expanded on the explanation of Eq. 4 in the resubmission.

  4. We have defined the acronym FCI in the resubmission.

  5. The calculation indeed does not involve an orbital optimization. We have included a mention to this in the body of the text. We have also added a reference to an example-calculation in our github repository as to not clutter the body of the paper with it.

  6. We have added to the description of Table 1 that the given complexities concern calculations without orbital optimization.

  7. The reviewer is correct to note that only a convergence with respect to an increasing seniority number is shown. We have adapted the manuscript appropriately.

  8. We have used the experimental value as given in the NIST database for the distances between the center of mass and the different atoms. We keep these fixed and indeed only variate the angle. We have added this to the manuscript.

  9. As requested, we have changed "bonding" to "bound".

  10. We indeed mean to say the tensor network calculations with full seniority. We have adapted the manuscript accordingly.

  11. The DOCI-optimized orbital for the Neon dimer are delocalized on one of the two atoms for larger separations. We have clarified this in the manuscript.

  12. We have elaborated further upon the computational complexity and the general usefulness of seniority-restricted tensor network calculations at the end of section 2.

In response to the second referee:

  1. We have commented upon the usage of DOCI-TNS in combination with orbital optimization in the conclusions.

  2. We have further commented on the entanglement present in the seniority-restricted wave functions at the end of section 2.

  3. We have elaborated on this in section 2.2.

  4. We have left out the mention of size-consistency of orbital optimized DOCI in the manuscript. This does not impinge the general message of the paragraph.

---

## Editorial Decision

published